# Cardiac Arrest: Can Technology Be the Solution?

**DOI:** 10.3390/jcm14030972

**Published:** 2025-02-03

**Authors:** Frédéric Lapostolle, Jean-Marc Agostinucci, Tomislav Petrovic, Anne-Laure Feral-Pierssens

**Affiliations:** 1SAMU 93, UF Recherche-Enseignement-Qualité, Université Paris 13, Sorbonne Paris Cité, Rue de Stalingrad, 93009 Bobigny, France; jeanmarc.agostinucci@croix-rouge.fr (J.-M.A.); anne-laure.feral-pierssens@aphp.fr (A.-L.F.-P.); 2Hôpital Avicenne, 125, Rue de Stalingrad, 93009 Bobigny, France; 3Inserm U942, Sorbonne Paris Cité, Rue de Stalingrad, 93009 Bobigny, France; tomislav.petrovic@aphp.fr

**Keywords:** cardiac arrest, basic life support, advanced life support, technology, education

## Abstract

Out-of-hospital cardiac arrest (OHCA) mortality remains alarmingly high in most countries. The majority of pharmacological attempts to improve outcomes have failed. Randomized trials have shown limited survival benefits with vasopressin, fibrinolysis, amiodarone, or lidocaine. Even the benefits of adrenaline remain a matter of debate. In this context, relying on technology may seem appealing. However, technological strategies have also yielded disappointing results. This is exemplified by automated external chest compression devices. When first introduced, theoretical models, animal studies, and early clinical trials suggested they could improve survival. Yet, randomized trials failed to confirm this benefit. Similarly, to date, extracorporeal membrane oxygenation (ECMO), therapeutic hypothermia, and primary angioplasty have demonstrated inconsistent survival advantage. Other technological innovations continue to be explored, such as artificial intelligence to improve the diagnosis of cardiac arrest during emergency calls, mobile applications to dispatch citizen responders to patients in cardiac arrest, geolocation of defibrillators, and even the delivery of defibrillators via drones. Nevertheless, it is clear that the focus and investment should prioritize the initial links in the chain of survival: early alerting, chest compressions, and defibrillation. Significant improvements in these critical steps can be achieved through the education of children. Modern technological tools must be leveraged to enhance this training by incorporating gamification and democratizing access to education. These strategies hold the potential to fundamentally improve the management of cardiac arrest.

## 1. The Success of Defibrillation

Efforts over the past decades to reduce out-of-hospital cardiac arrest mortality have largely failed [1]. Mortality rates remain alarmingly high worldwide, with survival rates often below 10% in most countries (Figure 1). While more than three-quarters of cardiac arrests occur at home, most often in the presence of witnesses, the diagnosis is rarely made, and the initiation of resuscitation efforts, such as external chest compressions, is infrequently undertaken [2,3]. It is widely stated that survival decreases by approximately 10% for every minute without intervention, making these initial moments critical for patient outcomes. The only significant success has been in chest compressions and, remarkably, in defibrillation. The landmark study conducted in Las Vegas casinos in 2000 established defibrillation as the cornerstone of out-of-hospital cardiac arrest management [4]. The training and widespread availability of automated external defibrillators (AEDs) among security officers staff led to impressive survival rates: 38% overall and as high as 59% for cases involving defibrillation. Notably, the most recent positive findings in the management of out-of-hospital cardiac arrest also pertain to advancements in defibrillation techniques [3,5]. Studies on anteroposterior defibrillation (used instead of or in addition to the conventional anterolateral approach) (Figure 2) and double defibrillation (delivering two successive shocks within one second) have shown significant improvements in outcomes [3]. Anteroposterior defibrillation and double defibrillation were associated with higher rates of return of spontaneous circulation (ROSC) and hospital survival compared to the traditional anterolateral approach [5]. Specifically, survival rates were 30% for double defibrillation (first defibrillation with conventional anterolateral pad placement and second defibrillation with anteroposterior pad placement), 22% for anteroposterior defibrillation, and 13% for conventional anterolateral defibrillation [3]. These recent findings highlight the central role of defibrillation in improving outcomes in out-of-hospital cardiac arrest and underscore the need for further research to optimize its application [6]. The rapid accessibility of defibrillators has become a major global challenge. Most public venues have since been equipped with these life-saving devices, with priority given to stadiums, train stations, and airports. At the turn of the century, airlines also took up this critical initiative, training their crews in basic life support and equipping their aircraft with defibrillators [7]. Early studies reported a survival rate of 50% among passengers who received defibrillation [8]. These findings also revealed that many patients benefitted from defibrillation using the aircraft’s defibrillator, sometimes even outside the aircraft itself.

## 2. The Failure of Pharmacological Solutions

### Adrenaline, Fibrinolysis, Anti-Arrhythmic Drugs, and Other Options

Adrenaline remains the main pharmacological agent in cardiac arrest management due to its dual cardiac (beta-stimulant) and vascular (alpha-stimulant) effects [9]. Nevertheless, its use in resuscitation is fraught with clinical complexity [10]. While adrenaline significantly increases the likelihood of return of spontaneous circulation during cardiopulmonary resuscitation (CPR)—with studies even suggesting higher doses (5 mg vs. 1 mg) may further enhance ROSC rates—its potent vasoconstrictive properties pose significant risks [9,11]. By maintaining coronary perfusion, adrenaline supports cardiac recovery, but it simultaneously exacerbates cerebral ischemia, often resulting in neurological deficits and poor long-term outcomes [12]. This dichotomy presents a critical challenge: how to balance the immediate life-saving benefits of adrenaline against its potentiality for severe neurological harm. Recent studies, including those by Hagihara et al. (2012) and Perkins et al. (2018), have underscored this paradox, calling for a more nuanced and individualized approach to adrenaline use in resuscitation [13,14]. Yet, in the absence of viable alternatives, adrenaline remains a mainstay of daily clinical practice in the advanced management of out-of-hospital cardiac arrest [1].

Attempts to improve outcomes with other pharmacological interventions have been largely disappointing. Vasopressin, proposed as an alternative to adrenaline, has shown no significant superiority [15]. The combination of nitrate derivatives with adrenaline, intended to mitigate cerebral vasoconstrictive damage, has likewise failed to demonstrate significant benefits. Similarly, calcium and corticosteroids have not demonstrated meaningful improvements [2,16].

In the same vein, fibrinolysis has proven ineffective [17]. Despite its theoretical appeal—addressing acute myocardial infarction or pulmonary embolism, which are common underlying causes of cardiac arrest—a randomized trial showed no significant survival benefit from early, out-of-hospital fibrinolysis. Anti-arrhythmic drugs have fared no better. No significant differences in survival were observed among patients with refractory ventricular fibrillation treated with amiodarone, lidocaine, or placebo (after direct, head-to-head comparison) [18]. Paradoxically, despite this lack of efficacy, lidocaine was added to resuscitation guidelines, a decision that defies logical reasoning in the absence of proven benefit [1]. This can be attributed to the fact that for clinicians who routinely manage cardiac arrest, particularly in patients with ventricular fibrillation—considered to have a relatively better prognosis—or those displaying signs of life, it is profoundly challenging to accept the absence of effective pharmacological interventions [19]. This reality underscores the catastrophic prognosis of out-of-hospital cardiac arrest and drives a natural inclination to turn to technology, hoping that where pharmacological solutions have failed, technological innovation might succeed. Before exploring which technologies could finally improve out-of-hospital cardiac arrest outcomes, it is important to examine the technologies that have already failed and understand the reasons for their shortcomings.

## 3. The Failure of Technological Solutions

### 3.1. The Mirage of Automated External Chest Compression Devices

Automated external chest compression devices were once anticipated to revolutionize the management and prognosis of out-of-hospital cardiac arrest (Figure 3) [4]. It had already been well established that chest compressions, alongside defibrillation, are key elements of cardiopulmonary resuscitation [20]. Coronary and cerebral perfusion, essential for the return of spontaneous circulation and favorable long-term neurological outcomes, are directly dependent on the quality and continuity of chest compressions [20]. Manual chest compressions, at best, provide only 30% of normal perfusion, and their interruption for even a few seconds results in a significant drop in aortic pressure to baseline levels (approximately to 10 mm Hg) [21,22]. Reestablishing effective perfusion requires several compressions to “prime the pump” again, underscoring the need for uninterrupted compressions [21]. These findings led to a prioritization of chest compressions, even at the expense of ventilation, during basic life support CPR [20,23]. Consequently, mouth-to-mouth ventilation has been omitted from basic life support guidelines. A randomized study demonstrated that this technique was not only ineffective (showing no significant difference in survival rates) but also associated with a 10% reduction in the initiation of chest compressions [24]. This decrease was attributed to reluctance or fear of performing mouth-to-mouth ventilation in the group where it was combined with chest compressions. Theoretically, this provided strong justification for the use of automated external chest compression devices. Operator fatigue and rapid degradation in chest compression quality further bolstered this argument [25]. Studies have shown that the quality of manual compressions deteriorates quickly—within a minute in some cases. Hightower et al. reported a decrease in correctly performed compressions from 92% in the first minute to just 39% by the third minute [26]. This rapid decline in performance, often unrecognized by the operator, added to the rationale for automation. Finally, adherence to guidelines—particularly the consistent and effective performance of chest compressions—has been shown to be an independent factor associated with the return of spontaneous circulation and improved survival (OR = 3.9; 95% CI: 1.1–14.0) in out-of-hospital cardiac arrest [27]. All these findings strongly supported the theoretical rationale for the use of automated external chest compression devices [12]. Even animal models supported these theoretical advantages, demonstrating improved hemodynamic outcomes with automated chest compressions compared to manual compressions, particularly when adrenaline was administered [28]. Automated devices delivered higher coronary and cerebral perfusion pressures and accelerated plasma peak adrenaline levels (90 vs. 150 s) compared to manual compressions [28]. Real-world studies on patients with out-of-hospital cardiac arrest also showed improved hemodynamics, including higher systolic, diastolic, and mean arterial pressures, and improved end-tidal carbon dioxide (EtCO_2_) levels [29]. All indicators pointed to automated chest compression devices as a promising tool to improve outcomes in out-of-hospital cardiac arrest. However, prospective randomized trials failed to confirm these expected benefits. A randomized study involving over 1000 patients using an automated load-distributed band chest compression device found no significant survival difference: 10% for manual CPR versus 6% for automated chest compressions [30]. Another trial, involving over 4000 patients and using another kind of automated chest compression device, with mechanical piston compressions, similarly showed no survival advantage: 11% for manual CPR versus 9% for automated chest compressions [31]. These findings ruled out technical flaws, shifting the failure to the conceptual level [32].

### 3.2. Hypothermia, Angioplasty, ECMO, and Intraosseous Venous Access

These results highlight a crucial lesson in out-of-hospital cardiac arrest management: no matter how robust the theoretical rationale, only randomized controlled trials can validate interventions, as confounding factors abound. This disappointment mirrors other technological interventions, such as precocious therapeutic hypothermia, which initially showed promise but failed to deliver clear survival benefits in randomized studies [33,34,35]. Finally, both the clinical benefit and level of evidence are low [36].

Likewise, angioplasty for patients who achieved ROSC was ineffective when no ST-segment elevation was present on the ECG, and its benefits remain unproven even in patients with ST-segment elevation [37,38]. Another example of unmet expectations is extracorporeal membrane oxygenation (ECMO). This strategy was initially implemented in the early 2000s for in-hospital cardiac arrest and was later extended to out-of-hospital cardiac arrest a few years afterward [39]. The concept of ECMO in cardiac arrest is to provide mechanical circulatory support in case of heart failure—cardiac arrest representing the ultimate manifestation of heart failure—enabling oxygenation and perfusion through extracorporeal circulation. The expansion of ECMO from in-hospital to out-of-hospital settings was designed to shorten the time to initiation of circulatory support. The timing of this intervention remains a critical factor: if ECMO is initiated too late, ischemic damage—particularly to the brain—negates its benefits, while initiating it too early risks unnecessary invasive procedures for patients who might have achieved ROSC with conventional resuscitation. The clinical dilemma surrounding ECMO revolves around its risk–benefit ratio and selection criteria. The initial French experience yielded disappointing early results, prompting the implementation of stricter patient selection criteria, focusing on younger patients (<70 years old), without significant comorbidities, witnessed arrests, persistent ventricular arrhythmias, short no-flow durations (time without chest compressions), and low-flow durations (time with reduced perfusion) of less than 5 and 20 min, respectively [40]. While this strategy reduced mortality, it inherently selected patients with intrinsically better prognosis [41]. Three randomized studies have examined the benefit of ECMO in cardiac arrest. Yannopoulos et al. reported (2020) significantly higher survival at 3 and 6 months in patients with refractory ventricular fibrillation treated with ECMO (43% vs. 0%) in a small-scale study, with 15 patients in each group [42]. However, subsequent randomized clinical studies—Belohlavek et al. (2022) and Suverein et al. (2023)—contradicted these preliminary findings [43,44]. In the former, which included 264 out-of-hospital cardiac arrest patients, and the latter, which included 134 such patients, no significant survival benefit was observed: 31% vs. 22% and 20% vs. 16%, respectively [43,44]. Two recent meta-analyses, incorporating both randomized and observational studies, suggested a potential benefit of ECMO for in-hospital cardiac arrest but remain inconclusive for out-of-hospital scenarios [45]. Importantly, ECMO is applicable to only a small subset of patients—1% in the French study and up to 6% in highly specialized centers [40,43]. Given the substantial human and material resources required, its cost-effectiveness, particularly in out-of-hospital settings, demands careful consideration.

Similarly, automated devices enabling intraosseous venous access as an alternative to peripheral venous access were initially anticipated to facilitate faster and easier drug administration [46]. The concept of intraosseous venous access is not new. It was first described and studied in the 1940s but remained underutilized due to the difficulty of needle placement [47]. However, the introduction of an automated device has dramatically transformed this practice [46]. Consequently, their use became increasingly widespread, despite a lack of evidence supporting these claimed benefits and regardless of their cost [48]. However, a recently published prospective randomized study once again demonstrated no significant clinical advantage [49]. The rate of return to spontaneous circulation (30% with intraosseous access vs. 29% with peripheral venous access) or the 30-day survival rate (12% vs. 10%) were not significantly improved. This failure reflects a broader tendency to expect technology to solve problems that healthcare professionals—whether physicians, nurses, or paramedics—are often capable of managing independently. Peripheral venous access, for example, can be achieved in 99% of cases after two or three attempts [50]. In complex, high-mortality situations, there is a natural inclination to expect more from technology than it can realistically deliver. Globally, the reliance on such invasive strategies underscores the clinician’s natural inclination to turn to technology when faced with the persistent failure of pharmacological and conventional interventions. This trend is not confined to medicine; similar discussions are found across various scientific disciplines. Many authors argue that the belief in salvation through (new) technology is a misguided hope—a fol espoir—that may ultimately deter efforts to seek appropriate human-driven solutions [51,52]. However, as with all therapeutic innovations, robust evidence from randomized trials remains the gold standard for determining their role in improving cardiac arrest outcomes. As illustrated with automated chest compression devices and ECMO, there are numerous similar examples. Nonetheless, it is crucial to critically and pragmatically evaluate such technologies before adopting them widely.

## 4. Which Technologies Could (Finally) Improve Cardiac Arrest Outcomes?

### 4.1. Artificial Intelligence

The past few decades have witnessed a technological revolution, profoundly transforming medicine through miniaturization, computerization, mechanization, and now artificial intelligence (AI) [53,54]. Despite high expectations, advancements in cardiac arrest management have struggled to translate into substantial improvements in survival outcomes. This article explores the technological interventions across the out-of-hospital management continuum, focusing on their potential impact on reducing ischemic brain injury—the primary determinant of cardiac arrest prognosis (Figure 4).

Prompt recognition of out-of-hospital cardiac arrest is critical for reducing no-flow duration, the period without effective circulation, which heavily influences outcomes. In most Western countries, cardiac emergencies begin with a call to the Emergency Medical System (EMS). However, misidentification of cardiac arrest delays treatment, while false-positive responses waste precious resources. AI has been hypothesized as a tool to improve dispatcher recognition of cardiac arrest. However, a randomized prospective trial showed no significant improvement when a machine learning alert system was added to dispatcher protocols (93% recognition with AI vs. 91% by human, without significant difference) [55]. The authors emphasized the need to enhance the specificity of machine learning algorithms.

Some authors have proposed using an artificial intelligence (AI) tool to determine the prognosis of a patient in cardiac arrest [56]. It is true that, to date, and in the absence of formal guidelines, the decision to terminate resuscitation is left to the physician’s discretion, without any objective criteria to assist them [57]. However, the availability of such a tool raises significant questions. The first is the issue of thresholds: at what survival probability should the decision be made to stop or continue resuscitation, and for how long? It is worth noting that physicians are generally quite accurate in predicting the death of critically ill patients. Many authors attribute this, at least in part, to a self-fulfilling prophecy, where knowledge of a poor prognosis may lead to less aggressive therapeutic efforts [58]. In the context of cardiac arrest, knowing that a patient has a low probability of survival (but at what threshold?) could lead to moderating or even stopping resuscitative efforts. Conversely, knowledge of a favorable prognosis (again, at what threshold?) might encourage prolonged therapeutic interventions that could be perceived as unreasonable. The concern that new technologies might dehumanize end-of-life care, including during cardiac arrest, is troubling. This is especially true given that studies have shown the importance of family presence during out-of-hospital cardiac arrest resuscitation. Families, when supported, benefit from being present and involved during these critical moments. Therefore, the use of AI in this context must be carefully considered to ensure it complements, rather than detracts from, humane and compassionate care.

Wearable devices, such as smartwatches, present a promising alternative for early detection. Having already demonstrated efficacy in atrial fibrillation detection, these devices could potentially alert EMS directly in the event of a cardiac arrest. The modeling of potential applications for this use has already been published [59].

### 4.2. Return to Defibrillation with the Good Samaritan

Once out-of-hospital cardiac arrest is identified, survival depends on the immediate initiation of life-saving measures: chest compressions and defibrillation. While the number of defibrillators has significantly increased in most Western countries over the past 25 years, several challenges remain [60]. Widespread access to automated external defibrillators (AEDs) currently faces three major challenges:

Availability: Many AEDs in public spaces are not accessible 24/7 [61]. In a recent study, we found that only 19% of automated external defibrillators (AEDs) were accessible 24/7. In France, fewer than one-third of the 500,000 installed AEDs are registered. This registration is crucial for ensuring true public access, a necessary step to improving cardiac arrest survival.Localization: Without mandatory registration, AED locations are not always accurately documented or updated. Such geolocation—still missing in many countries—must also be continuously updated and made immediately accessible, accurately and without errors.Maintenance: Studies have shown that some AEDs are nonfunctional due to issues such as defective electrodes or depleted batteries [61]. A recent French study found that 60% of automated external defibrillators (AEDs) were suspected of not being immediately functional, most commonly due to maintenance failures [62].

To address the two critical elements of cardiopulmonary resuscitation—early chest compressions and early defibrillation—coordination and accessibility are essential. Therefore, citizen responder systems have been implemented in several countries (Figure 5) [63,64]. These systems use mobile apps to alert registered volunteers near the out-of-hospital cardiac arrest site and guide them to the patient, often providing AED location information if available [65]. Early results show increased rates of bystander CPR and improved 30-day survival rates [63]. The current challenges include standardizing the use of such applications, increasing the number of registered volunteers (particularly in isolated and underserved areas), and integrating these applications with dispatch center software to ensure that rescuers are mobilized immediately, regardless of the platform where they are registered.

The concept of drone-delivered AEDs has been proposed to further expedite defibrillation [66]. While modeling studies show promise, logistical challenges, high costs, and complex deployment strategies limit feasibility [67,68]. Operational uncertainties, including drone localization and activation protocols, remain unresolved. Despite the enthusiasm, randomized trials have yet to confirm the real-world impact of this approach, leading the authors of a recent review to conclude succinctly: “there is significant potential” [69,70].

A more credible advancement lies in improving the detection of ventricular fibrillation by (semi-automatic) defibrillators [71]. Currently, cardiac rhythm analysis requires interruptions in chest compressions—a practice whose harmful effects have been extensively documented [72]. However, with the integration of artificial intelligence, defibrillators are expected to soon perform rhythm analysis without necessitating any pause in chest compressions [73].

Although less influential on survival compared to prehospital interventions, advanced in-hospital tools are under development. For instance, smart glasses could guide nonmedical responders in performing high-quality CPR [74]. Hemodynamic monitoring to assess compression efficacy via femoral vessel examination with ultrasound or cerebral perfusion monitoring with transcranial Doppler or near-infrared spectroscopy (NIRS) has been proposed [75,76,77]. However, these methods are not yet practical for routine use due to complexity and lack of proven efficacy. To date, the stringent requirements of randomized trials have often dashed even the most legitimate hopes of improving survival rates in cardiac arrest cases. 

A recent review highlights a discrepancy between the perceived importance of the rescue chain components and their actual impact on outcomes (Figure 4A) [78]. Early links of the chain—alerting, chest compressions, and defibrillation—are the most critical determinants of survival that would deserve to be (significantly) expanded (Figure 4B). The majority of efforts should be focused on this critical step of life support. The authors also highlighted another major discrepancy: the distribution of research funding across the various links in the chain of survival. Unsurprisingly, the “medical” link—primarily hospital-based—receives the majority of investments, despite the fact that this link is far from being the most critical in determining outcomes. The neurological prognosis, in particular, is likely already largely determined before the patients reach the hospital after the return of spontaneous circulation (ROSC), given the significant impact of no-flow and low-flow durations on outcomes [19].

Countries investing in public education, particularly within school systems, have achieved significant improvements in bystander CPR rates and survival outcomes [79]. Over the past decades, significant efforts have been made to simplify basic life support (BLS) guidelines in order to increase their implementation. Diagnostic strategies have been streamlined: pulse checking has been abandoned, and the new rule is now ‘No, No, Go’. If there is no response and no breathing, the next step is clear: start chest compressions. The guidelines for chest compressions have also been simplified. Hands should be placed ‘in the middle of the chest’. Mouth-to-mouth ventilation has been abandoned, paving the way for the inclusion of children in BLS training programs. Research has shown that chest compressions can be effectively performed by children aged 12 and older. However, training in alerting emergency services, using automated external defibrillators, and initiating chest compressions should begin as early as possible. Incorporating gamification, virtual reality, and AI-based tools into school curricula could offer cost-effective, scalable solutions to train entire generations in life-saving techniques [80,81]. Technological innovations offer exciting opportunities to improve the prognosis of out-of-hospital cardiac arrest. However, their adoption must focus on practical implementation, evidence-based evaluation, and equitable access. Nevertheless, prioritizing education and the optimization of early interventions through innovative tools remains the most effective strategy to enhance survival outcomes.

## 5. Could Technology Primarily Serve Education?

New strategies to improve cardiac arrest outcomes should not be implemented in out-of-hospital settings without robust evidence from randomized studies. However, that alone is not sufficient. While cost-effectiveness is an important endpoint, it is also not enough. The strategy must target a substantial number or proportion of patients who could benefit from its implementation. In public health terms, prioritizing a strategy such as extracorporeal membrane oxygenation (ECMO)—which, regardless of its effectiveness, applies to only a small percentage of cardiac arrest patients—while neglecting public education efforts aimed at significantly increasing the identification of cardiac arrests, the initiation of chest compressions, and the use of defibrillators, would be unethical.

Children’s education is arguably the most impactful lever to improve survival rates from out-of-hospital cardiac arrest [82]. Evidence supports this: children trained using video-based instruction alone (without hands-on practice) achieve performance levels comparable to those trained traditionally with mannequins [83]. Such findings have spurred the creation of numerous training programs across various countries. A framework for these educational efforts was outlined in a landmark publication [84]. However, while many innovative models and approaches exist, robust evaluations of their outcomes remain scarce. Scaling these initiatives effectively will likely require the proliferation of diverse programs tailored to local needs and capacities. Furthermore, developing the most effective tools necessitates collaboration with specialized engineers and experts in emerging technologies. Imagine deploying resources for designing an educational game to teach life-saving techniques comparable to those allocated for developing a blockbuster video game. With such investment, we could revolutionize education in basic life support—and potentially win the battle against cardiac arrest mortality. By blending gamification, immersive learning experiences, and advanced analytics, these tools could not only engage but also empower learners of all ages. The dream of a universally educated population, capable of performing life-saving interventions, may finally be within reach.

## 6. Conclusions

Addressing the management of out-of-hospital cardiac arrest often feels like recounting a series of disappointments and unmet expectations. Physicians, perhaps mistakenly, have placed significant hope—and continue to do so—in technological innovations. This inclination is not unique to medicine; it is a natural response when faced with a critical, life-threatening condition such as cardiac arrest, which is associated with a staggering mortality rate, exceeding 90% in most cases. Unfortunately, aside from defibrillation, no intervention has proven capable of substantially altering this grim prognosis.

However, leveraging technology to educate every child in a given age group may represent the most promising strategy to finally achieve meaningful progress. By prioritizing large-scale, technology-driven education, we have a tangible opportunity to reduce the mortality associated with out-of-hospital cardiac arrest and improve outcomes on a global scale.

## Figures and Tables

**Figure 1 jcm-14-00972-f001:**
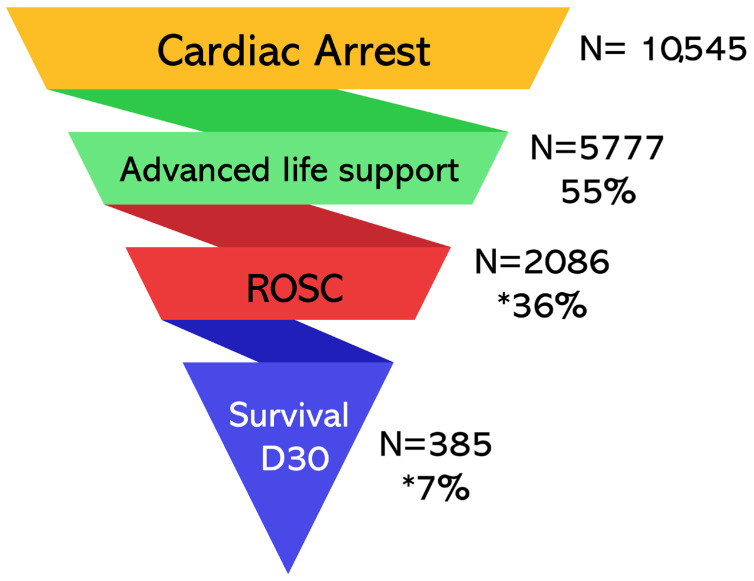
Data on the Management of Out-of-Hospital Cardiac Arrest by Mobile Intensive Care Unit (MICU—SAMU-93) in the great Paris area, France (2013–2024). Advanced life support refers to the implementation of medical resuscitation by the French MICU. ROSC: return of spontaneous circulation. * Calculated based on the number of medical advanced life supports performed.

**Figure 2 jcm-14-00972-f002:**
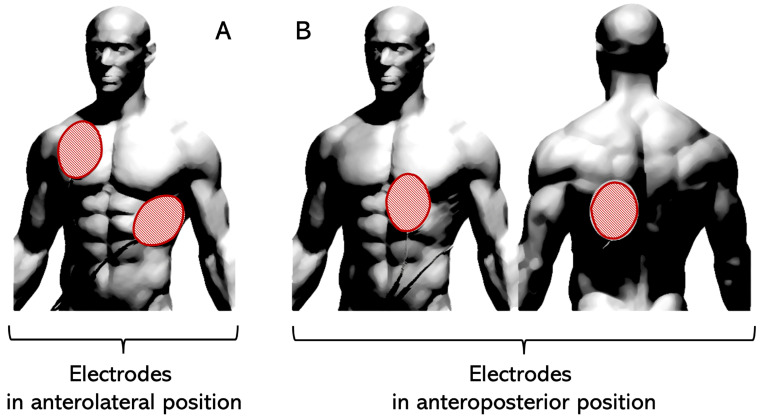
Defibrillation electrodes’ placement: (**A**) historical anterolateral placement; (**B**) anteroposterior placement that has been shown to improve defibrillation efficacy [3].

**Figure 3 jcm-14-00972-f003:**
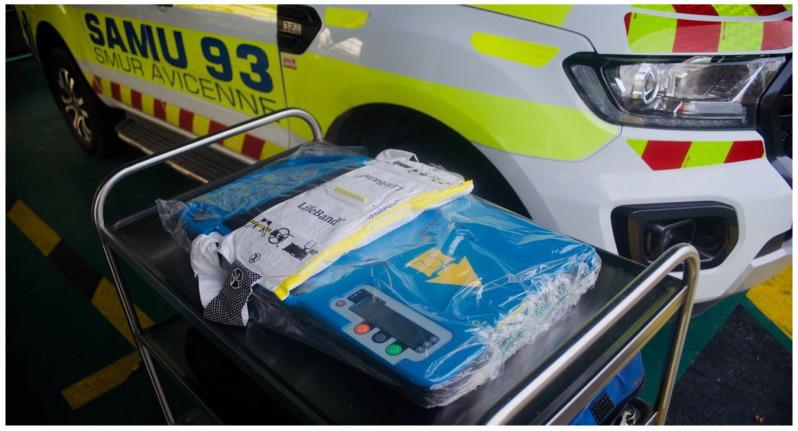
Automated load-distributed band chest compression device. © FLAPO.

**Figure 4 jcm-14-00972-f004:**
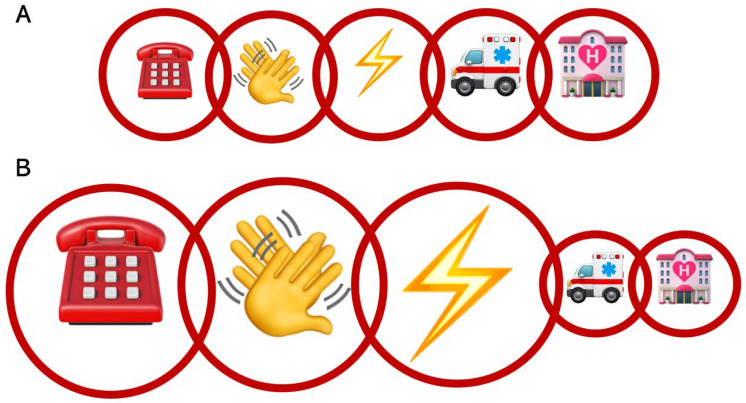
Chain of survival: alert, chest compressions, defibrillation, advanced care = medical intervention, hospitalization. (**A**) Classical representation. (**B**) Representation based on prognostic impact.

**Figure 5 jcm-14-00972-f005:**
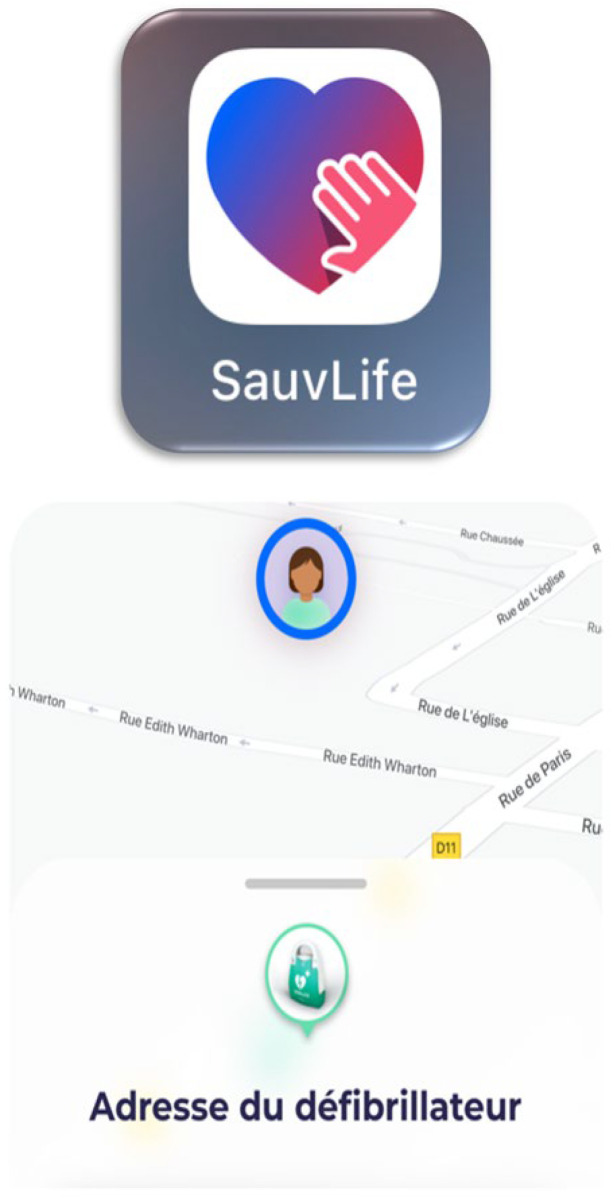
‘Citizen’ application: the location of the cardiac arrest victim and a nearby defibrillator is sent to the registered citizen rescuer via the application. Non-English terms in figure are defibrillator address.

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
