# Peer review of "Cardiac Arrest: Can Technology Be the Solution?"

_jcm, 2025, doi:10.3390/jcm14030972_

Round 1

Reviewer 1 Report

Comments and Suggestions for Authors

The manuscript "Cardiac Arrest: Can Technology Be the Solution?" explores the challenges and opportunities associated with technological advancements in the management of out-of-hospital cardiac arrest (OHCA). The authors present a comprehensive perspective on the role of technology, highlighting past successes and failures while proposing future directions for innovation. This manuscript addresses a critically important topic in resuscitation science, considering the persistently low OHCA survival rates worldwide.

The manuscript’s strength lies in its structured approach, beginning with an overview of key developments, such as defibrillation, and progressing to the challenges of pharmacological and technological solutions. It effectively synthesizes a wide range of studies and real-world data, offering a balanced assessment of the potential and limitations of various interventions.

The discussion on defibrillation is particularly insightful. By reviewing advancements such as anteroposterior defibrillation and double defibrillation, the authors emphasize the significant impact of these techniques on the return of spontaneous circulation (ROSC) and survival outcomes. Recent evidence strongly supports this section, underscoring the importance of optimizing defibrillation techniques in enhancing OHCA outcomes.

The manuscript also critically evaluates the role of pharmacological interventions, emphasizing the limitations of adrenaline and other agents such as vasopressin, nitrates, and antiarrhythmic drugs. The authors provide a nuanced discussion of the risks and benefits of adrenaline, highlighting its dual effects on cardiac recovery and cerebral ischemia. This balanced perspective reflects the complexity of resuscitation pharmacology and the ongoing need for individualized approaches.

The authors then transition to a thorough review of technological interventions, such as automated chest compression devices, extracorporeal membrane oxygenation (ECMO), and intraosseous venous access systems. The discussion is candid about the disappointing results from large-scale randomized trials for these technologies, which failed to deliver the anticipated survival benefits. This analysis underscores the importance of evidence-based practice and the need for randomized controlled trials to validate the efficacy of new interventions.

Despite these challenges, the manuscript remains forward-looking, exploring emerging technologies with the potential to revolutionize OHCA management. For instance, the authors discuss the integration of artificial intelligence (AI) in early recognition of cardiac arrest, wearable devices for real-time monitoring, and innovations in defibrillator design that enable rhythm analysis without interrupting chest compressions. These insights highlight the potential of technology to address critical gaps in the chain of survival.

The manuscript also emphasizes the importance of public education and community engagement in improving bystander response rates. The discussion on gamification, virtual reality training, and AI-driven educational tools is particularly compelling, offering practical strategies for equipping broader populations with life-saving skills. This focus on education as a cost-effective and scalable solution aligns well with global public health priorities.

However, there are areas for improvement. Firstly, while the manuscript provides an excellent overview of past and current technologies, it could benefit from a clearer framework for evaluating future innovations. Establishing criteria for assessing the feasibility, cost-effectiveness, and scalability of new technologies would enhance the practical utility of the manuscript. Additionally, the discussion on AI and wearable devices could be expanded to include potential ethical and logistical challenges, such as data privacy concerns and disparities in access to advanced technologies.

Author Response

See File

Reviewer 2 Report

Comments and Suggestions for Authors

Thank you for opportunity to review a perspective article "Cardiac Arrest: Can Technology Be the Solution?".

This is well written article which focus on all aspects where technology can improve the menagment and outcome after OHCA. The authors discussed technology that has not been verified and individual weak points that may be improved in the future. Possible improvements to current systems such as AED are also being sought. An article deserves for publication but it needs minor changes.

Strenghts: robust and detailed review, easy to read

Weakness: lack of title subsections that could further improve readability,

Please improve Figure 4 - it should be more intuitive for reader.

Consider citation good narrative review about predicting models: Spadafora L, Biondi-Zoccai G, Bernardi M. Out-of-hospital cardiac arrest: predict and then protect!. EBioMedicine. 2023;90:104517.

Author Response

See File

Reviewer 3 Report

Comments and Suggestions for Authors

The authors reviewed current technologies used in cardiopulmonary resuscitation of out of hospital cardiac arrest. The technologies include defibrillation position, chest compression, pharmacological agent, ECMO and et al. The authors carefully discussed the success and failure of these technologies and provided their perspective about how to improve CPR quality to increase the survival rate.

Major concern:

According to the 2024 Heart Disease and Stroke Statistics Update Fact Sheet published by American Heart Association, the main reason for the very low survival rate less than 10% is due to the incidence of cardiac arrest primarily occurred at home (72.1%) and these patients, especially those single persons, cannot receive in-time CPR rescue. Importantly, the 4-5 minutes after cardiac arrest incidence is the gold time to reach successful resuscitation. However, it is very challenging for patients who have cardiac arrest at home to receive CPR during this time period especially for those who are alone at home.

Therefore, without addressing this key issue, advancing CPR technologies alone is impossible to improve the overall survival rate. With the new emerging technologies, I would suggest the authors to provide their insight about how to use the innovative technologies to help the at-home patients receive the rapid EMS service. Having a new section to specifically discuss the development of new technologies and/or strategies specifically for at-home patients will strength this article.

Issues:

1.      An abstract is needed.

2.      Line 11-12: Please explain why the survival rate is very low.

3.      Line 20: It is confusing to understand what is in the bracket.

4.      Line 25: survival rates were 30% for double defibrillation – Please specify the defibrillation pads’ location for double defibrillation.

5.      Line 37: It is better to replace ‘cornerstone with ‘primary’ or ‘main’

6.      Line 61: Please edit the content in the bracket

7.      Line 107: Please use the full name of EtCo2: end-tidal carbon dioxide since it is used once.

8.      Line 249 and 251: Figure 5A and 5B should be Figure 4A and 4B, in my understanding.

Author Response

See File

Round 2

Reviewer 3 Report

Comments and Suggestions for Authors

Great work!